# Adverse Events in Targeted Therapy for Unresectable Hepatocellular Carcinoma Predict Clinical Outcomes

**DOI:** 10.3390/cancers16183150

**Published:** 2024-09-14

**Authors:** Kenji Imai, Koji Takai, Masashi Aiba, Shinji Unome, Takao Miwa, Tatsunori Hanai, Atsushi Suetsugu, Masahito Shimizu

**Affiliations:** Department of Gastroenterology/Internal Medicine, Graduate School of Medicine, Gifu University, 1-1 Yanagido, Gifu 501-1194, Japan; takai.koji.t2@f.gifu-u.ac.jp (K.T.); aiba.masashi.v4@f.gifu-u.ac.jp (M.A.); unome-shinji@hotmail.com (S.U.); hanai.tatsunori.p8@f.gifu-u.ac.jp (T.H.); suetsugu.atsushi.e2@f.gifu-u.ac.jp (A.S.); shimizu.masahito.j1@f.gifu-u.ac.jp (M.S.)

**Keywords:** hepatocellular carcinoma, targeted therapy, adverse event, overall survival, progression-free survival

## Abstract

**Simple Summary:**

The most common adverse events (AEs) that occurred in response to targeted therapy for hepatocellular carcinoma were appetite loss (adverse event grade 0/1/2/3 = 97/23/55/12), general fatigue (102/31/44/6), hypertension (120/6/40/17), hand-foot syndrome (HFS) (135/21/24/3), proteinuria (140/13/16/14), and hypothyroidism (148/12/23/0). Among these, appetite loss and general fatigue negatively affect overall survival (OS) and progression-free survival (PFS). Increasing AE grades of hypertension, proteinuria, and hypothyroidism were associated with better OS, whereas hypertension, HFS, and hypothyroidism were associated with better PFS.

**Abstract:**

To assess the impact of adverse event (AE) severity, caused by targeted therapy, on overall survival (OS) and progression-free survival (PFS) in patients with unresectable hepatocellular carcinoma (HCC), a total of 183 patients with HCC treated with atezolizumab plus bevacizumab (40), lenvatinib (57), sorafenib (79), cabozantinib (3), ramucirumab (3), and regorafenib (1) were included in this study. Age-, AFP-, and ALBI score-adjusted hazard ratios (HRs) of AE grades 1 to 3 versus grade 0 for OS and PFS were calculated using Cox proportional hazards models. The linear trend of the HRs was assessed by calculating the *p* values for this trend. The most common AEs were appetite loss (AE grade 0/1/2/3 = 97/23/55/12), general fatigue (102/31/44/6), hypertension (120/6/40/17), hand-foot syndrome (HFS) (135/21/24/3), proteinuria (140/13/16/14), and hypothyroidism (148/12/23/0). The adjusted HRs for OS of these AEs were 0.532–1.450–2.361 (*p* for trend 0.037), 1.057–1.691–3.364 (*p* for trend 0.004), 1.176–0.686–0.281 (*p* for trend 0.002), 0.639–0.759–1.820 (*p* for trend 0.462), 1.030–0.959–0.147 (*p* for trend 0.011), and 0.697–0.609 (*p* for trend 0.119), respectively. Those for PFS of the corresponding AEs were 0.592–1.073–2.811 (*p* for trend 0.255), 1.161–1.282–4.324 (*p* for trend 0.03), 0.965–0.781–0.655 (*p* for trend 0.095), 0.737–0.623–2.147 (*p* for trend 0.153), 1.061–0.832–0.800 (*p* for trend 0.391), and 1.412–0.560 (*p* for trend 0.081), respectively. Appetite loss and general fatigue negatively affected clinical outcomes, whereas hypertension, HFS, proteinuria, and hypothyroidism had positive effects.

## 1. Introduction

Hepatocellular carcinoma (HCC) is a common malignancy [1,2]. Clinical guidelines recommend surgical resection or radiofrequency ablation (RFA) for curable HCC [3,4,5]. However, HCC is often diagnosed only after its progression to an incurable stage [6]. Furthermore, because HCCs recur easily and repeatedly at sites other than the primary site via multicenter carcinogenesis [1], most HCCs progress to an incurable state. Transcatheter arterial chemoembolization (TACE) or systemic targeted therapy is recommended for these patients [3,4,5]. Recently, systemic targeted therapy has gained increased importance in treating unresectable HCC owing to both dramatic advances in systemic targeted therapy [7,8,9,10,11,12] and concerns about the negative impact of TACE on the liver functional reserve [13].

The SHARP trial confirmed that sorafenib (SOR), an oral multikinase inhibitor of the vascular endothelial growth factor receptor (VEGFR), prolonged survival by nearly three months compared with placebo (7). Other VEGFR inhibitors, including regorafenib (REG) [8], lenvatinib (LEN) [9], cabozantinib (CAB) [10], and ramucirumab (RAM) [11], have also shown survival benefits in patients with HCC. Furthermore, the IMbrave150 trial found that the combination of atezolizumab (Ate), a programmed death-ligand 1-targeting antibody, and bevacizumab (Bev), a VEGFR-targeting monoclonal antibody, resulted in better overall survival (OS) and progression-free survival (PFS) than SOR [12]. Currently, Ate + Bev is recommended as the first-line therapy for unresectable HCC, with other drugs reserved for later-line therapy [14].

However, various adverse events (AEs) have become more apparent with the increasing use of targeted therapies. General fatigue and digestive symptoms, such as appetite loss and diarrhea, are the most common AEs. Hypertension, hand-foot syndrome (HFS), proteinuria, and hypothyroidism, which may result from anti-VEGF activity, are also common targeted therapies [7,8,9,10,11,12]. Most AEs that occur in response to targeted therapy for HCC can negatively affect the health-related quality of life of patients, leading to poor clinical outcomes. However, several studies have indicated that some AEs are associated with improved patient outcomes [15]. For example, patients treated with SOR who experienced dermatological AEs within the first 60 days of treatment showed a significantly increased time to progression [16]. Patients who developed hypertension after SOR treatment and proteinuria after Ate + Bev treatment also showed a significant improvement in OS [17,18]. However, other studies reported contradictory results [19]. Therefore, some adverse events arising during targeted therapy for hepatocellular carcinoma may improve clinical outcomes, whereas others may exacerbate them.

To improve the prognosis of patients with unresectable HCC, it is necessary to accurately assess the current status of AEs and their impact on clinical outcomes, and to implement the most effective measures for their prevention and management. This study aimed to determine the prevalence of AEs in response to targeted therapy (Ate + Bev, LEN, SOR, CAB, RAM, and REG) for unresectable HCC and to assess their impact on clinical outcomes, including OS and PFS.

## 2. Materials and Methods

### 2.1. Patients and Treatment Strategy

All 183 patients with HCC treated with Ate + Bev (n = 40), LEN (n = 57), SOR (n = 79), CAB (n = 3), RAM (n = 3), or REG (n = 1) between May 2009 and December 2022 at Gifu University Hospital were enrolled in this study. Drug selection was conducted according to the most up-to-date HCC treatment guidelines available at the time [5,20]. Specifically, before introducing immune checkpoint inhibitors (ICIs), SOR or LEN was used as the first-line therapy, while other agents were reserved as subsequent treatment lines. After the advent of ICIs, Ate + Bev was used as the first-line therapy, with the other agents being used in the later lines of therapy. In cases where ICIs were contraindicated, SOR or LEN was used as the first-line therapy. SOR was administered at a dose of 400 mg/day (half of the recommended dose) because of concerns regarding serious AEs, whereas the other drugs were administered at the recommended dose. Dose modifications were not permitted in the Ate + Bev group, though were allowed in the other groups according to the severity of the AEs. When intolerable AEs or tumor growth were observed, a change in the treatment strategy was considered in accordance with the guidelines [14]. If a therapeutic agent was changed during the observation period, the patient was assigned to the initial treatment group (e.g., a patient who commenced with SOR and subsequently transitioned to REG was classified under the SOR group). Monitoring was continued until the patient’s death or the observation period termination. The response of each patient to therapy was assessed using dynamic computed tomography (CT), according to the modified RECICT [21]. AEs were assessed according to the *Common Terminology Criteria for Adverse Events*, version 5.0. OS and PFS were defined as the interval from the date of drug introduction to the date of death or progressive disease (PD), respectively.

Patients enrolled in this study were offered the opportunity to opt out after receiving information about the study. The study design, including the consent procedure, was approved by the ethics committee of Gifu University School of Medicine on 10 January 2024 (ethical protocol code: 2023–239).

### 2.2. Statistical Analyses

OS and PFS were estimated using the Kaplan–Meier method, and the differences between the strata of the AE grade (0–3) of appetite loss, general fatigue, hypertension, HFS, proteinuria, and hypothyroidism were evaluated using a log-rank test, and multiple comparisons were adjusted using Bonferroni correction. Hazard ratios (HRs) and 95% confidence intervals (CIs) for AE grades 1–3 versus grade 0 were calculated for overall survival (OS) and progression-free survival (PFS) using Cox proportional hazards models. Adjusted HRs and 95% CIs for age, alpha-fetoprotein (AFP) level, and ALBI scores are also provided. The linear trend of the HRs was assessed by calculating the *p* values for this trend. Statistical significance was set at *p* < 0.05. All statistical analyses were performed using the R software ver. 4.2.2 (R Foundation for Statistical Computing, Vienna, Austria; http://www.R-project.org/ [accessed on 4 April 2024]).

## 3. Results

### 3.1. Baseline Clinical Characteristics and Clinical Course of the Enrolled Patients

The baseline clinical characteristics of the enrolled patients (147 men; median age, 73 years) immediately before the initiation of targeted therapy are shown in Table 1. At the start of the current study, 168 patients had already received other treatments (hepatectomy/RFA/TACE/radiation/other targeted therapy = 89/56/132/47/39), whereas 94 patients received further treatment after discontinuing the targeted therapy (hepatectomy/RFA/TACE/radiation/other targeted therapy = 0/2/39/13/41). Targeted therapy was initiated in 144 patients as first-line treatment, 30 patients as second-line treatment, 8 patients as third-line treatment, and 1 patient as fourth-line treatment. The number of first- and later-line cases for AB, LEN, SOR, CAB, RAM, and REG were 30/10, 40/17, 73/6, 0/3, 1/2, and 0/1, respectively (Appendix A). The median treatment duration for the enrolled patients was 5.6 months.

### 3.2. Treatment Response and Frequency of AEs in the Enrolled Patients

The complete response (CR), partial response (PR), stable disease (SD), progressive disease (PD), and not evaluable (NE) for all treatments were 11, 30, 60, 74, and 8 patients, respectively (Table 1), and the objective response and disease control rates were 22.4% and 55.2%, respectively. The OS rates at 1, 2, and 3 years, and the median OS for all patients were 64.4%, 36.4%, 21.0%, and 17.2 months, respectively. The PFS rates were 35.7%, 15.5%, 7.9%, and 7.2 months, respectively. During the analysis period, 112 patients died, and 152 patients experienced PD. There was no significant difference in OS between the first- and later-line treatment groups (*p* = 0.0688); however, there was a significant difference in PFS (*p* = 0.0478) (Appendix A).

The most frequent AEs were appetite loss (AE grade 0/1/2/3 = 97/23/55/12), general fatigue (AE grade 0/1/2/3 = 102/31/44/6), hypertension (AE grade 0/1/2/3 = 120/6/40/17), HFS (AE grade 0/1/2/3 = 135/21/24/3), proteinuria (AE grade 0/1/2/3 = 140/13/16/14), and hypothyroidism (AE grade 0/1/2 = 148/12/23). The frequency and severity of AEs associated with each drug are listed in Table 2. Patients who received targeted therapy as later-line therapy had significantly more severe AEs of proteinuria (*p* = 0.004) and hypothyroidism (*p* = 0.027) than those who received the first-line therapy (Appendix A).

### 3.3. Impact of the Severity of AEs on OS and PFS

The cumulative incidence of OS divided by the AE grades of appetite loss, general fatigue, hypertension, HFS, proteinuria, and hypothyroidism is shown in Figure 1, with *p* values evaluated using the log-rank test. These results show that appetite loss and general fatigue negatively impacted OS, whereas hypertension, proteinuria, and hypothyroidism positively impacted OS.

Figure 2 shows the non-adjusted and adjusted HRs of AE grades 1–3 compared with those of grade 0 for OS. The adjusted HRs for the OS of appetite loss, general fatigue, hypertension, and proteinuria were 0.532, 1.450, and 2.361 (*p* for trend 0.037); 1.057, 1.691, and 3.364 (*p* for trend 0.004); 1.176, 0.686, and 0.281 (*p* for trend 0.002); and 1.030, 0.959, and 0.147 (*p* for trend 0.011), respectively, which were significantly different. These results also clearly demonstrated that OS significantly worsened with increasing AE grades for appetite loss and general fatigue, whereas it improved with increasing AE grades for hypertension and proteinuria.

The cumulative incidence of PFS according to the AE grades of appetite loss, general fatigue, hypertension, HFS, proteinuria, and hypothyroidism is shown in Figure 3. Similar to OS, appetite loss and general fatigue were significantly associated with worse PFS, whereas these improved in the groups that developed HFS and hypothyroidism.

Figure 4 shows the non-adjusted and adjusted HRs of AE grades 1–3 compared with those of grade 0 for PFS. Increasing the AE grades of general fatigue (*p* = 0.043) were significantly associated with worse PFS in the non-adjusted HRs, whereas hypertension (*p* = 0.011), HFS (*p* = 0.011), and hypothyroidism (*p* = 0.022) were significantly associated with better PFS. Among them, general fatigue (*p* for trend = 0.03) was significantly associated with worse PFS in the adjusted HRs.

## 4. Discussion

The results of the present study showed that increasing AE grades for hypertension, proteinuria, and hypothyroidism were associated with better OS, whereas hypertension, HFS, and hypothyroidism were associated with better PFS. The exact mechanisms underlying these AEs and their association with improved clinical outcomes remain unclear. However, the anti-VEGF activity of all agents used in this study may be associated with improved clinical outcomes and the development of these AEs [15,16,22,23,24].

Angiogenesis is a crucial step in the expansive growth and metastasis of tumors because tumors that increase in diameter beyond several millimeters require vascular networks to supply oxygen and nutrients for further growth [25]. VEGF, a major molecule associated with angiogenesis, has garnered attention as a potential therapeutic target for cancer. For HCC treatment, Bev (a VEGF-targeting monoclonal antibody) has been approved in combination with Ate (an immune checkpoint inhibitor). RAM (a monoclonal antibody against VEGFR-2) and other multikinase inhibitors such as LEN, SOR, CAB, and REG have been approved as monotherapies [7,8,9,10,11,12].

Targeted therapies that include these agents are known to induce specific AEs such as hypertension, proteinuria, hypothyroidism, and HFS. Hypertension is thought to be caused by the inhibition of the VEGF-mediated upregulation of NO synthase [26]. These agents can also induce proteinuria by inhibiting VEGFR-2 in glomerular capillary endothelial cells, thereby damaging endothelial cells, basement membranes, and podocytes, and contributing to the filtration barrier of the renal glomerulus [24]. Inhibition of VEGF signaling can impair thyroid function through the regression of capillaries around the thyroid follicles [24]. Therefore, the severity of these AEs may correlate with the efficacy of the treatment, as they may represent the extent to which VEGF signaling in the tumor can be suppressed [18]. Therapeutic efficacy and tolerability should be considered along with appropriate AE management. However, given the positive impact of VEGF-related AEs on clinical outcomes, caution should be exercised while reducing the dose or changing the treatment strategy for patients with HCC who experience these AEs while receiving targeted therapy, including these agents.

In the present study, hypertension, the third most common AE, significantly improved the OS and PFS. Therefore, regular BP monitoring during the first few months of targeted therapy is useful and, if hypertension develops, it should be controlled with appropriate antihypertensive medications to continue targeted therapy [15]. Proteinuria, the fifth most common AE, significantly improved OS in this study. This finding is consistent with those of previous studies [18,27]; however, other studies have shown contradictory results [19]. The only way to manage AE is to interrupt or reduce treatment; however, it is important to maintain a high relative dose intensity to improve survival [28]. Therefore, treatment doses should be strictly adjusted according to the severity of proteinuria. Hypothyroidism, the sixth most common AE, significantly improved OS and PFS in univariate analysis. To the best of our knowledge, no previous studies have demonstrated an association between VEGF inhibitor-induced hypothyroidism and improved outcomes in patients with HCC. The management of AE for hypothyroidism involves regular monitoring of thyroid function and the introduction of thyroid replacement therapy if detected. Clinicians should also understand that hypothyroidism can induce AEs such as general fatigue and appetite loss, which can negatively impact OS and PFS [15]. HFS, the fourth most common AE, significantly improved PFS in the univariate analysis, which is consistent with the findings of a previous study [16]. However, if HFS progresses to AE grade 3, both OS and PFS deteriorate. To avoid and minimize HFS, emollients containing 10% urea three times a day are recommended [15]. It is also important to educate patients on wearing cotton socks and gloves to protect against sun and cold exposure and unnecessary friction, which can worsen HFS [15].

Consistent with previous reports [15], this study showed that the progression of general fatigue and appetite loss, two of the most common AEs of targeted therapy for unresectable HCC, worsened clinical outcomes, including OS and PFS. As appetite loss and general fatigue worsen, the risk of treatment interruption and the onset of cachexia or sarcopenia, both recognized prognostic factors, increases [29,30], thereby contributing to unfavorable clinical outcomes. Moreover, this study demonstrated that both appetite loss and general fatigue significantly elevate HRs when categorized as AE grade 2 or higher for OS and grade 3 or higher for PFS. Notably, among the 44 patients who experiencedAE grade ≥ 2 general fatigue, 36 (81.8%) also experienced AE grade ≥ 2 appetite loss. These findings suggest that maintaining these adverse events within AE grade 1 is associated with a more favorable prognosis. Therefore, appropriate prevention and management of these AEs during targeted therapy are of paramount importance for improving OS and PFS.

To prevent general fatigue at an early stage, it is important to educate patients regarding the possibility of developing treatment-related symptoms [15]. Patients experiencing fatigue should be advised to incorporate rest periods into their daily schedules [15]. Other pathological conditions that may cause fatigue, such as low testosterone levels, pain, sleep dysfunction, depression, hypothyroidism, and anemia, should also be considered and treated appropriately [15,31]. Monitoring a patient’s appetite and weight during targeted therapy is important for the early detection of AEs associated with appetite loss. High-calorie diets and nutritional supplements are recommended for patients experiencing appetite loss [15]. Adequate nutritional therapy is also essential to address nutritional disorders in chronic liver disease, which underlies many HCCs [32]. In addition, these AEs may be caused not only by the side effects of targeted therapy but also by HCC itself. Therefore, if these AEs are intolerable or if the therapeutic effect is weakened, reducing the dose of targeted therapy or changing the strategy for HCC should be considered.

All drugs are approved at fixed doses. However, the large interindividual variability in pharmacokinetics and pharmacodynamics (PK/PD) remains an unsolved clinical issue [33]. Recently, therapeutic drug monitoring (TDM) strategies have proven effective in balancing therapeutic efficacy and managing AEs when using multikinase inhibitors in cancer treatment [33]. Regarding lenvatinib, there are reports that the weekend-off method is effective in mitigating adverse events [34]. Carnitine deficiency has also been implicated in the onset of general fatigue [35]. Therefore, TDM strategies, the weekend-off method, and the administration of carnitine supplements may potentially contribute to the prevention of appetite loss and general fatigue.

This study had several limitations. First, this was a retrospective single-center study with a relatively small sample size. Second, the anticancer drugs used in this study ranged from monoclonal antibodies to multikinase inhibitors, although all drugs possessed anti-VEGF activity. It is well known that the adverse event profiles differ between monoclonal antibody therapies and multikinase inhibitors. For example, monoclonal antibodies that specifically block vascular endothelial VEGFR do not lead to HFS [23]. Furthermore, the AEs observed in the patients treated with Ate + Bev could not be definitively attributed to either drug alone. Third, the study included patients who received these drugs as first- and late-line treatments. This study revealed that patients who received targeted therapy as a later-line therapy had significantly shorter PFS and more severe AEs of proteinuria and hypothyroidism. Thus, the frequency of AEs and their impact on clinical outcomes may depend on the targeted therapy line. In addition, 94 patients (51.4%) received some form of post-treatment after targeted therapy was discontinued; therefore, the effect of post-treatment on OS could not be excluded. To address these issues, it is essential to gather more cases of HCC treated with targeted therapy from multiple institutions, and prospectively examine the correlation between adverse events and clinical outcomes.

## 5. Conclusions

In targeted therapy for unresectable HCC, the progression of AE grades of general fatigue and appetite loss is associated with poor OS and PFS. In contrast, AEs thought to be related to the inhibition of VEGF signaling, such as increasing AE grades of hypertension, proteinuria, and hypothyroidism, were associated with better OS, whereas hypertension, HFS, and hypothyroidism were associated with better PFS. Appropriate assessment, prevention, and management of AEs are essential to improve the clinical outcomes of patients with HCC treated with targeted therapy.

## Figures and Tables

**Figure 1 cancers-16-03150-f001:**
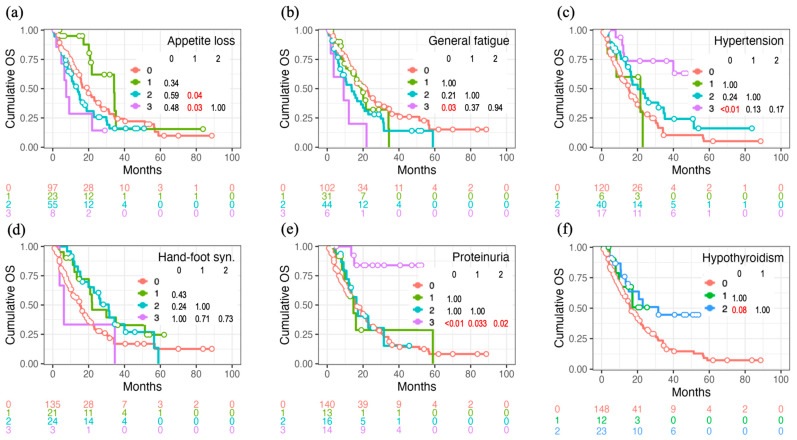
Overall survival (OS) curves for each grade of targeted therapy-induced adverse event. The differences between the survival curves (**a**–**f**) were assessed using the log-rank test, with multiple comparisons adjusted via Bonferroni correction.

**Figure 2 cancers-16-03150-f002:**
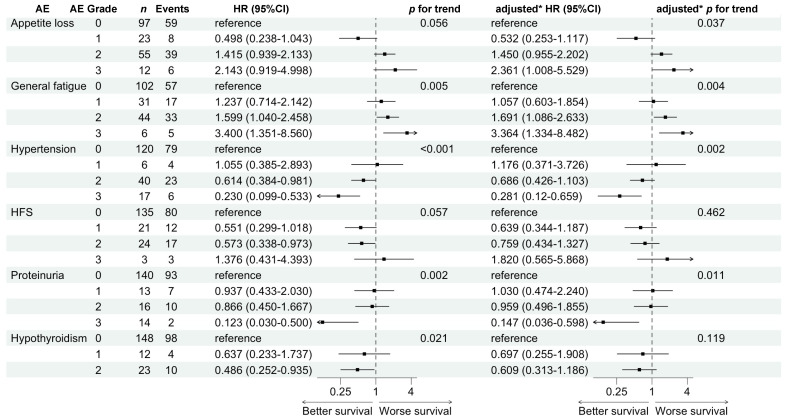
Hazard ratios for overall survival comparing adverse event grades 1 to 3 with 0. * Adjusted for age, alpha-fetoprotein, and ALBI score.

**Figure 3 cancers-16-03150-f003:**
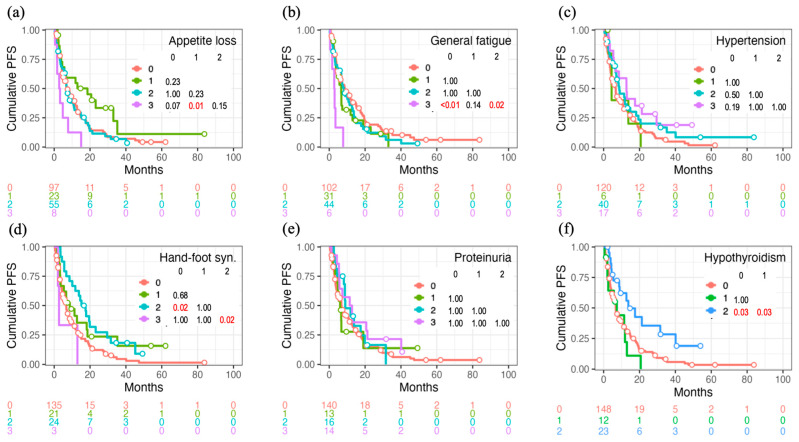
Progression-free survival (PFS) curves for each grade of targeted therapy-induced adverse event. The differences between the survival curves (**a**–**f**) were assessed using the log-rank test, with multiple comparisons adjusted via Bonferroni correction.

**Figure 4 cancers-16-03150-f004:**
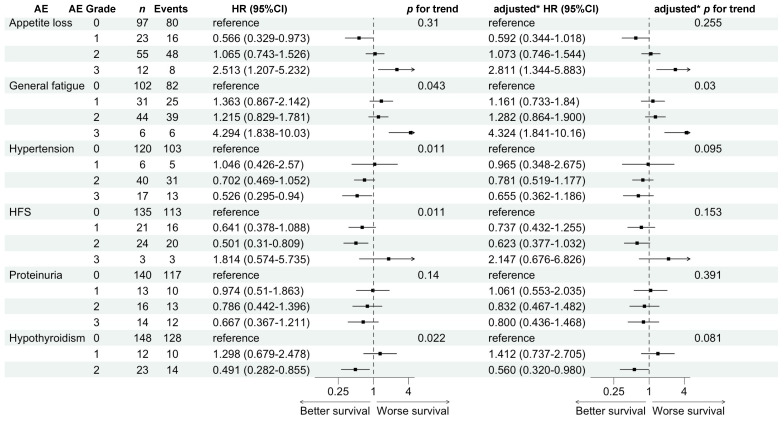
Hazard ratios for progression-free survival comparing adverse event grades 1 to 3 with 0. * Adjusted for age, alpha-fetoprotein, and ALBI score.

**Table 1 cancers-16-03150-t001:** Baseline demographic and clinical characteristics of the enrolled patients.

Variables	n = 183
Age (years)	73 (66–79)
Sex (male/female)	147/36
Etiology (HBV/HCV/others)	40/60/83
Drug (AB/LEN/SOR/CAB/RAM/REG)	40/57/79/3/3/1
Target therapy line (1/2/3/4)	144/30/8/1
BCLC stage (A/B/C)	12/67/102
Child–Pugh score (5/6/7/8/9)	103/62/13/3/1
ALBI score	−2.435 (−2.778–−2.115)
ECOG PS (0/1/2/3)	140/32/10/1
AFP (×10^3^ ng/mL)	100.8 (8.1–1682.8)
PIVKA-II (×10^3^ mAU/mL)	583.0 (62.5–5334.5)
Pre-treatment (no/any/hepatectomy/RFA/TACE/RT/other targeted therapy)	15/168/89/56/132/47/39
Post-treatment (no/any/hepatectomy/RFA/TACE/RT/other targeted therapy)	89/94/0/2/39/13/41
Best response (CR/PR/SD/PD/NE)	11/30/60/74/8
Treatment duration (month)	5.6 (1.6–12.1)

Continuous covariates are presented as median (interquartile range). HBV, hepatitis B virus; HCV, hepatitis C virus; AB, atezolizumab plus bevacizumab; LEN, lenvatinib; SOR, sorafenib; CAB, cabozantinib; RAM, ramucirumab; REG, regorafenib; BCLC stage, Barcelona Clinic Liver Cancer stage; ECOG, Eastern Cooperative Oncology Group; PS, performance status; RFA, radiofrequency ablation; TACE, transcatheter arterial chemoembolization; RT, radiation therapy; CR, complete response; PR, partial response; SD, stable disease; PD, progressive disease; NE, not evaluable.

**Table 2 cancers-16-03150-t002:** Adverse events during targeted therapy.

	All	AB	LEN	SOR	CAB	RAM	REG
Appetite loss (G0/1/2/3)	97/23/55/12	27/7/5/1	24/3/28/2	41/12/22/4	1/1/0/1	3/0/0/0	1/0/0/0
General fatigue (G0/1/2/3)	102/31/44/6	25/12/3/0	25/2/27/3	48/14/14/3	2/1/0/0	2/1/0/0	0/1/0/0
Hypertension (G0/1/2/3)	120/6/40/17	25/2/9/4	34/2/14/7	55/2/16/6	3/0/0/0	3/0/0/0	0/0/1/0
Hand-foot syndrome (G0/1/2/3)	135/21/24/3	40/0/0/0	40/7/10/0	49/14/13/3	2/0/1/0	3/0/0/0	1/0/0/0
Proteinuria (G0/1/2/3)	140/13/16/14	21/8/5/6	39/4/9/5	76/1/1/1	2/0/0/1	1/0/1/1	1/0/0/0
Hypothyroidism (G0/1/2)	148/12/23	27/9/4	40/0/17	77/1/1	2/0/1	1/2/0	1/0/0

AB, atezolizumab/bevacizumab; LEN, lenvatinib; SOR, sorafenib; CAB, cabozantinib; RAM, ramcirumab; REG, regorafenib.

## Data Availability

The data presented in this study are available upon request from the corresponding author.

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
