# Peer review of "Adverse Events in Targeted Therapy for Unresectable Hepatocellular Carcinoma Predict Clinical Outcomes"

_cancers, 2024, doi:10.3390/cancers16183150_

Round 1

Reviewer 1 Report

Comments and Suggestions for Authors

Overall Evaluation

This study aimed to evaluate the impact of the severity of chemotherapy adverse events (AEs) on overall survival (OS) and progression-free survival (PFS) in patients with unresectable hepatocellular carcinoma (HCC). A total of 183 patients who were admitted between May 2009 and December 2022 were enrolled in the study, and the side effects of drug treatment were analyzed (40 patients were treated with Atezolizumab + Bevacizumab; 57 with Lenvatinib; 79 with Sorafenib; 3 with Cabozantinib; 3 with Ramucirumab; 1 with Regorafenib). The side effects were analyzed using a Cox proportional hazards model. The final results showed that appetite loss and general fatigue negatively affected clinical outcomes, whereas hypertension, hand-foot syndrome (HFS), proteinuria, and hypothyroidism positively affected clinical outcomes. Currently, several studies have reported on the relationship between drug side effects and liver cancer, so the innovation of this study is generally limited.

Recommendations:

Title

1.       Atezolizumab, bevacizumab, Lenvatinib, sorafenib, cabozantinib, ramucirumab, regorafenib are strictly targeted therapies, not chemotherapy, so the word "chemotherapy" in the title should be replaced with "targeted therapy".

Introduction:

1.       In line 70, the author mentioned that “Therefore, the impact of AEs on clinical outcomes, such as OS and PFS, in patients with unresectable HCC remains unclear”. However, there have been many studies on the relationship between AEs and liver cancer, and this description is inappropriate.

Materials and Methods:

1.       The author should give clear inclusion and exclusion criteria in this session.

2.       If the patients reached intolerable AEs or tumor growth, were these patients still included in the current study?

Results:

1.       In line 126 is the sentence “The complete response (CR), partial response (PR), stable disease (SD), and PD for all treatments were 11, 30, 60, and 74 patients, respectively”. In total, there were 175 cases. So, where the left 9 cases go?

2.       In the Figure 4, the author should analyze the results between p for trend and adjusted p for trend was different?

Discussion:

1.       In the first sentence, the author wrote “The results of the present study showed that the progression of the AE grade in hypertension, proteinuria, hypothyroidism and HFS tended to be associated with better clinical outcomes”. It is better to demonstrate in the clear description, such as OS and DFS instead of clinical outcomes.

2.       It is not necessary to discuss a lot about the exact mechanisms in a clinical study.

To make the results more readable, the author should visualize the relationship between AEs and DFS and OS in a table. It is hard for the readers to make these reulsts clear. For me, the first time when I read, I was totally confused.

Author Response

We are pleased that in the overall comments this reviewer found our study is of interest. We also thank this reviewer’s constructive comments which were most helpful to improve our manuscript. We accordingly revised the manuscript as follows.

  1. Atezolizumab, bevacizumab, Lenvatinib, sorafenib, cabozantinib, ramucirumab, regorafenib are strictly targeted therapies, not chemotherapy, so the word "chemotherapy" in the title should be replaced with "targeted therapy".

Following your advice, I have replaced all instances of “chemotherapy” with “targeted therapy” throughout the text.

  1. In line 70, the author mentioned that “Therefore, the impact of AEs on clinical outcomes, such as OS and PFS, in patients with unresectable HCC remains unclear”. However, there have been many studies on the relationship between AEs and liver cancer, and this description is inappropriate.

In accordance with your suggestion, I have revised the relevant sections as follows; Some adverse events arising during targeted therapy for hepatocellular carcinoma may improve clinical outcomes, while others may exacerbate them. (lines 71-72)

  1. The author should give clear inclusion and exclusion criteria in this session.
  2. If the patients reached intolerable AEs or tumor growth, were these patients still included in the current study?

In this study, all cases in which targeted therapy was initiated for hepatocellular carcinoma during the observation period (May 2009 to December 2022) were included, and This study continued until death or the end of the observation period, even if there was a switch to another treatment including another targeted therapy. I have added this description to the Materials and Methods section. (lines 81 and lines 90-91)

  1. In line 126 is the sentence “The complete response (CR), partial response (PR), stable disease (SD), and PD for all treatments were 11, 30, 60, and 74 patients, respectively”. In total, there were 175 cases. So, where the left 9 cases go?

The remaining 8 cases were unable to be evaluated for therapeutic response due to the lack of imaging examinations during the observation period. This information has been added to Table 1 and the main text. (lines 123, 129 and 132)

  1. In the Figure 4, the author should analyze the results between p for trend and adjusted p for trend was different?

Figure 4 shows the non-adjusted and adjusted HRs AE of grades 1–3 compared with those of grade 0 for PFS. Increasing AE grades of general fatigue (p for trend = 0.043) were significantly associated with worse PFS in non-adjusted HRs, whereas those of hypertension (p for trend = 0.011), HFS (p for trend = 0.011), and hypothyroidism (p for trend = 0.022) were significantly associated with better PFS. Among them, general fatigue (p for trend = 0.03) were significantly associated with worse PFS in adjusted HRs. We have added this description concerning Figure 4 in the Results section. (lines 175-180)

  1. In the first sentence, the author wrote “The results of the present study showed that the progression of the AE grade in hypertension, proteinuria, hypothyroidism and HFS tended to be associated with better clinical outcomes”. It is better to demonstrate in the clear description, such as OS and DFS instead of clinical outcomes.

This study clearly demonstrated that increasing AE grades for hypertension, proteinuria, and hypothyroidism were associated with better OS and those of hypertension, HFS, and hypothyroidism were associated with better PFS. For readers to understand this result easily, we have revised the relevant section more clearly in the Simple Summary, Discussion, and Conclusions sections. (lines15-17, 185-187, and 289-291)

  1. It is not necessary to discuss a lot about the exact mechanisms in a clinical study. To make the results more readable, the author should visualize the relationship between AEs and DFS and OS in a table. It is hard for the readers to make these reulsts clear. For me, the first time when I read, I was totally confused.

In accordance with your suggestion, I have partially removed the description regarding the mechanism by which VEGF promotes cancer progression. However, I have retained most of the discussion on the mechanisms by which each adverse event occurs and how these events influence prognosis, as I consider this to be a critical component of the paper. Additionally, in response to the feedback that the study’s results were difficult to understand, I have revised the text as previously mentioned to make it clearer (lines15-17, 185-187, and 289-291) and have also modified the Graphical Abstract to ensure that readers can easily grasp the results immediately.

The comments provided by this reviewer has been invaluable and made our revised manuscript more logical and comprehensible. I extend my deepest gratitude for your support.

Reviewer 2 Report

Comments and Suggestions for Authors

The authors reported the relationship between adverse events and OS in HCC patients who received systemic chemotherapy. Moreover, appetite loss and general fatigue negatively impacted clinical outcomes such as OS and PFS. Although the reviewer agrees with these results, I have found some concerns that, once addressed, will improve the manuscript.

Major

(1)  The author analyzed AEs including all drugs (Atez/Bev, SORA, LEN, CABO, REGO, RAM) in this study. I think the incidence of AEs varies for each drug. The author should be shown all AEs for each drug separately.

(2)  In this study, 39 patients received other chemotherapy in pre-treatment. In the later-line therapy, OS and PFS were shorter than 1st-line therapy. Moreover, the incidence of AEs has also increased in the later-line therapy. How does the author address these biases?.

(3)  In the comparison of RFS and OS according to each AE grade in this study, the author should use the methods of Bonferroni.. etc.

Minor

(1)  The author should describe the “number at risk” in Figures 1 and 3

Author Response

We are pleased that in the overall comments this reviewer found our study is of interest. We also thank this reviewer’s constructive comments which were most helpful to improve our manuscript. We accordingly revised the manuscript as follows.

Major

  1. The author analyzed AEs including all drugs (Atez/Bev, SORA, LEN, CABO, REGO, RAM) in this study. I think the incidence of AEs varies for each drug. The author should be shown all AEs for each drug separately. 

 Following your advice, we have created a new Table 2, which details the adverse events for each drug separately. (Lines 143-144 and 147-149)

  1. In this study, 39 patients received other chemotherapy in pre-treatment. In the later-line therapy, OS and PFS were shorter than 1st-line therapy. Moreover, the incidence of AEs has also increased in the later-line therapy. How does the author address these biases?

In accordance with your suggestions, we have added information regarding the target therapy line of the patients to Table 1 (lines 119-121, and 123). Additionally, we analyzed the differences in OS and PFS between first-line and later-line therapies, as well as the differences in AEs. (Supplementary Figure 1 and Supplementary Table 1). Consequently, there was no significant difference in OS between first- and later-line groups (p = 0.0688), but significant difference in PFS (p= 0.0478). Patients who introduced targeted therapy as later-line had significantly more severe AEs of proteinuria (p = 0.004) and hypothyroidism (p = 0.027) than those who introduced as first-line. This information has been added to the Results section and the Discussion section as one of the limitations in this study. (lines 137-139, 144-146, and 278-281)

  1. In the comparison of RFS and OS according to each AE grade in this study, the author should use the methods of Bonferroni.. etc.

 Following your advice, we conducted multiple comparisons between AE grades for OS and PFS using Bonferroni correction and added the p values to Figure 1 and Figure 3. (lines 103-104, 156, and 173).

Minor

  1. The author should describe the “number at risk” in Figures 1 and 3

Following your advice, number at risk have been added to Figure1 and 3.

The comments provided by this reviewer has been invaluable, particularly in enhancing the clarity of the study’s key findings, making them more comprehensible to readers. I am profoundly grateful for your kindness.

Reviewer 3 Report

Comments and Suggestions for Authors

The authors investigated the prevalence of adverse events (AEs) in response to systemic therapy for hepatocellular carcinoma (HCC) and assessed their impact on clinical efficacy. In general, this is a well-written paper that present interesting data. However, there are several drawbacks in the manuscript.

Major

1)     The word 'chemotherapy” has been used throughout the manuscript.  As systemic therapy for HCC does not include cytotoxic agents, the term 'chemotherapy' is inappropriate.

2)     Since the authors stated that it is necessary to implement the most effective measures for their prevention and management in the introduction, further discussion is needed. Specifically, in the case of lenvatinib treatment for HCC, the weekend-off method has been reported to avoid AEs (PMID: 32325921). Additionally, carnitine insufficiency has been associated with the development of fatigue during lenvatinib therapy, indicating the possibility of levocarnitine supplementation (PMID: 32126131).

3)     The severity of AEs during tyrosine kinase inhibitor treatment depends on dose adjustment. I think it would be useful to discuss the development of AEs in terms of the pharmacokinetic-pharmacodynamic theory.

4)     It is recommended that a post-hoc analysis be conducted following the log-rank test among the four groups in Figures 1 and 2.

5)     The authors found that appetite loss and general fatigue were associated with a worse PFS and OS. It would be helpful to discuss the reasons for this in terms of the development mechanism.

6)     Determining the impact of anorexia and fatigue severity on OS and PFS in this cohort is of considerable importance. Furthermore, it would be beneficial to explore the relationship between anorexia and fatigue.

7)     One limitation of this study is that it did not distinguish between the first and subsequent lines of treatment with anti-VEGF agents. Sequential use of these agents has been associated with an increased risk of adverse events.  Further analysis of systemic therapy in 1st line only is also needed.

Minor

1) Line 59: The phrase 'associated with HCC' is inappropriate. 

2) Line 136: Figure 2 seems to have been placed before Figure 1, which causes confusion.

3) There are too few line breaks in the discussion section. It is recommended that suitable line breaks be inserted.

4) Lines 227-228: Please provide an appropriate citation.

Author Response

We are pleased that in the overall comments this reviewer found our study is of interest. We also thank this reviewer’s constructive comments which were most helpful to improve our manuscript. We accordingly revised the manuscript as follows.

Major

  1. The word 'chemotherapy” has been used throughout the manuscript.  As systemic therapy for HCC does not include cytotoxic agents, the term 'chemotherapy' is inappropriate.

Following your advice, I have replaced all instances of “chemotherapy” with “targeted therapy” throughout the text.

  1. Since the authors stated that it is necessary to implement the most effective measures for their prevention and management in the introduction, further discussion is needed. Specifically, in the case of lenvatinib treatment for HCC, the weekend-off method has been reported to avoid AEs (PMID: 32325921). Additionally, carnitine insufficiency has been associated with the development of fatigue during lenvatinib therapy, indicating the possibility of levocarnitine supplementation (PMID: 32126131).

As you suggested, the weekend-off method and levocarnitine supplementation appear to be highly effective strategies for preventing AEs and mitigating their severity. Accordingly, I have added new citation (#33 and #34) and included these strategies as one of the methods for preventing and appropriately managing AEs in the Discussion section. (lines 265-269)

  1. The severity of AEs during tyrosine kinase inhibitor treatment depends on dose adjustment. I think it would be useful to discuss the development of AEs in terms of the pharmacokinetic-pharmacodynamic theory.

As you suggested, optimizing dosage based on PK/PD theory also appears to be an effective strategy for preventing adverse events and maximizing therapeutic efficacy. With adding literature supporting the effectiveness of therapeutic drug monitoring (TDM) (#32), I have included TDM based on PK/PD theory as one of the methods for preventing and appropriately managing AEs. (lines 261-265 and 267-269)

  1. It is recommended that a post-hoc analysis be conducted following the log-rank test among the four groups in Figures 1 and 2.

Following your advice, we conducted multiple comparisons between AE grades for OS and PFS using Bonferroni correction and added the p values to Figure 1 and Figure 3. (lines 103-104, 156, and 173).

  1. The authors found that appetite loss and general fatigue were associated with a worse PFS and OS. It would be helpful to discuss the reasons for this in terms of the development mechanism.

We think that as appetite loss and general fatigue worsen, the risk of treatment interruption and the onset of cachexia or sarcopenia—both recognized prognostic factors—increase, thereby contributing to unfavorable clinical outcomes. We have added this description in the Discussion section with new references (#28 and #29, lines 237-240)

  1. Determining the impact of anorexia and fatigue severity on OS and PFS in this cohort is of considerable importance. Furthermore, it would be beneficial to explore the relationship between anorexia and fatigue.

This study demonstrated that both appetite loss and general fatigue significantly elevate HRs when categorized as AE grade 2 or higher for OS, and as grade 3 or higher for PFS. Notably, among the 44 patients who experienced AE grade ≥ 2 general fatigue, 36 (81.8%) also experienced AE grade ≥ 2 appetite loss. These findings suggest that keeping the AEs of appetite loss and general fatigue within AE grade 1 is associated with a more favorable prognosis. We have added this description in the Discussion section. (lines 240-245)

  1. One limitation of this study is that it did not distinguish between the first and subsequent lines of treatment with anti-VEGF agents. Sequential use of these agents has been associated with an increased risk of adverse events.  Further analysis of systemic therapy in 1st line only is also needed.

In accordance with your suggestions, we have added information regarding the target therapy line of the patients to Table 1 (lines 119-121, and 123). Additionally, we analyzed the differences in OS and PFS between first-line and later-line therapies, as well as the differences in AEs. (Supplementary Figure 1 and Supplementary Table 1). Consequently, there was no significant difference in OS between first- and later-line groups (p = 0.0688), but significant difference in PFS (p= 0.0478). Patients who introduced targeted therapy as later-line had significantly more severe AEs of proteinuria (p = 0.004) and hypothyroidism (p = 0.027) than those who introduced as first-line. This information has been added to the Results section and the Discussion section as one of the limitations in this study. (lines 137-139, 143-146, and 278-281)

Minor 

  1. Line 59: The phrase 'associated with HCC' is inappropriate.  
  2. Line 136: Figure 2 seems to have been placed before Figure 1, which causes confusion. 
  3. There are too few line breaks in the discussion section. It is recommended that suitable line breaks be inserted. 
  4. Lines 227-228: Please provide an appropriate citation.

All concerns have been appropriately addressed and corrected.

The comments provided by this reviewer has been invaluable, particularly in enhancing the clarity of the study’s key findings, making them more comprehensible to readers. I am profoundly grateful for your kindness.

Round 2

Reviewer 1 Report

Comments and Suggestions for Authors

Recommendations:

1.       In line 88-89, the author mentioned that the treatment strategy was considered in accordance with guidelines (Systemic treatment of hepatocellular carcinoma: An EASL position paper) if AEs or tumor growth were observed. I just wonder if the patient who changed treatment strategy within the research time, for example from Sorafenib to Regorafenib, were still in Sorafenib group or changed into Regorafenib or even excluded from the study?

2.       According to the guidelines (Systemic treatment of hepatocellular carcinoma: An EASL position paper), the treatment algorithm for HCC candidates to systemic therapy recommend first-line and second-line therapy. However, the author addressed in the manuscript there were total fourth-line treatment. So, I just wonder what is the third-line and fourth-line, please give the detail and reference.

Author Response

Response to Reviewer #1

We are pleased that in the overall comments this reviewer found our study is of interest. We also thank this reviewer’s constructive comments which were most helpful to improve our manuscript. We accordingly revised the manuscript as follows.

  1. In line 88-89, the author mentioned that the treatment strategy was considered in accordance with guidelines (Systemic treatment of hepatocellular carcinoma: An EASL position paper) if AEs or tumor growth were observed. I just wonder if the patient who changed treatment strategy within the research time, for example from Sorafenib to Regorafenib, were still in Sorafenib group or changed into Regorafenib or even excluded from the study?

If a therapeutic agent was changed during the observation period, the patient was assigned to the initial treatment group (e.g., a patient who commenced with SOR and subsequently transitioned to REG was classified under the SOR group). Monitoring was continued until the patient’s death or the observation period termination.

The description regarding the protocol for drug modification has been revised as outlined above to enhance clarity for the reader. (lines 94-98)

  1. According to the guidelines (Systemic treatment of hepatocellular carcinoma: An EASL position paper), the treatment algorithm for HCC candidates to systemic therapy recommend first-line and second-line therapy. However, the author addressed in the manuscript there were total fourth-line treatment. So, I just wonder what is the third-line and fourth-line, please give the detail and reference.

Drug selection was conducted according to the most up-to-date HCC treatment guidelines available at the time. Specifically, before introducing immune checkpoint inhibitors (ICIs), SOR or LEN was used as the first-line therapy, while other agents were reserved as subsequent treatment lines. After the advent of ICIs, Ate+Bev was used as the first-line therapy, with other agents being used in later lines of therapy. In cases where ICIs were contraindicated, SOR or LEN was used as the first-line therapy.  We have added this information concerning drug selection for HCC with new references (#20 and #21) which are HCC treatment guidelines before and after the advent of ICIs. (lines 83-89)

Reviewer 2 Report

Comments and Suggestions for Authors

The author has made appropriate revisions.

Author Response

Response to Reviewer #2

We are pleased that in the overall comments this reviewer found our study is of interest. We also thank this reviewer’s constructive comments which were most helpful to improve our manuscript. We accordingly revised the manuscript as follows.

The author has made appropriate revisions.

Thank you for evaluating the revised manuscript. There are no further modifications to be made in response to this reviewer.

Reviewer 3 Report

Comments and Suggestions for Authors

In line with the reviewer's comment, the revised manuscript has been improved.

Author Response

Response to Reviewer #3

We are pleased that in the overall comments this reviewer found our study is of interest. We also thank this reviewer’s constructive comments which were most helpful to improve our manuscript. We accordingly revised the manuscript as follows.

In line with the reviewer's comment, the revised manuscript has been improved.

Thank you for evaluating the revised manuscript. There are no further modifications to be made in response to this reviewer.

Round 3

Reviewer 1 Report

Comments and Suggestions for Authors

According to the most up-to-date HCC treatment guidelines, systemic therapy is categorized into first-line systemic therapy and second-line and subsequent systemic therapies. In the manuscript, the authors mentioned that targeted therapy was initiated in 144 patients as first-line treatment, 30 as second-line, eight as third-line, and one as fourth-line treatment (line 126). Could the authors please clarify which drugs were used in each treatment line? Alternatively, it may be more effective to divide the treatments into two groups: first-line and second-line/subsequent therapies.

Author Response

In accordance with your suggestion, we have specified the number of cases for each anticancer drug and treatment line used in this study. The number of first- and later-line cases for AB, LEN, SOR, CAB, RAM, and REG were 30/10, 40/17, 73/6, 0/3, 1/2, and 0/1, respectively. These results have been summarized in Supplementary Table 1 and described in the Recult section. (lines 128-130)

Round 4

Reviewer 1 Report

Comments and Suggestions for Authors

no more suggestions